# Factors which Influence Risk Taking and the Evolution of Social-Identity in Stroke Narratives: A Thematic Synthesis

**DOI:** 10.3390/bs10020046

**Published:** 2020-01-31

**Authors:** Richard Higgs, Andrew Soundy

**Affiliations:** School of Sport, Exercise and Rehabilitation Sciences, University of Birmingham, Birmingham B15 2TT, UK; richyhiggs@live.co.uk

**Keywords:** stroke, narratives, social identity, rehabilitation, experiences, perceptions

## Abstract

**Background:** The disruption of a stroke can impact an individual’s sense of social identity. A comprehensive review is required to understand the factors and processes that influence changes in social identity following a stroke. **Aims:** To undertake a review of literature to discover a process of social identity evolution for people with stroke and identify the factors which influence it. **Methods:** A meta-ethnographic approach to review was undertaken and a subtle realist viewpoint was assumed. Studies were included if they documented experiences and perceptions relating to stroke. Eight electronic databases were searched from January 2009 until January 2019. Quality assessment and synthesis techniques were applied. **Findings:** Out of the 18 papers included, a total of 251 (141/251, 56% male, 109/251, 43% female, 1/251, 0.4% undisclosed) individuals were included within the synthesis. The evolution of social-identity model was developed and identified with five key stages to represent a process that individuals with stroke can experience. Factors which influence the process were identified and direct implications for clinical practice are given. **Conclusion:** This review has highlighted the major themes within the evolution of social identity and management strategies for risk taking to achieve a desired future. Further research is required to consider how these findings may be tested in clinical practice.

## 1. Introduction

Within the UK, there are more than 100,000 cases of stroke recorded each year [1]. The average age of onset has decreased within higher income countries across the past 30 years [2] and one in four strokes affect the working aged population [3]. Currently, there are 1.2 million stroke survivors in the UK with estimated survival rates for people aged > 45 years rising by 123% due to reduced mortality rates [4]. Despite this, the effects to the individual in terms of emotional stress, altered lifestyle and strain on relationships as a result of decreased physical and psychosocial capability can lead to a poor quality of life. The disruption and devastation of a stroke to a person’s lifestyle will impact daily routine, hobbies, roles and relationships within their life, leaving them with a distorted sense of social identity [5]. Social identity has been defined as “*an individual’s knowledge that he or she belongs to certain social groups together with some emotional and value significance to him or her or this group members*” [6].

Studies have found a positive association between a strong social identity and positive mental well-being. For instance, activities which give individuals a sense of purpose and enhance social identity have been associated with increased independence and decreased depression and anxiety [7]. However, many individuals who have suffered a stroke can struggle to achieve a desired social identity [8]. Social identity is formed by reviewing the past, and foreseeing a potential future by evaluating one’s present circumstance. Clinicians are required to understand the processes and factors which influence an individual’s social identity [9]. Given the above, further research is needed which considers how social identity is impacted by the onset of a stroke.

Past reviews highlighting the impact of a stroke on social identity have done so by focusing on the impact and change in roles following stroke caused by changes in social support and family relationships [10,11]. Other past reviews [12,13] have recognised the importance and impact of a stroke on an individual’s social identity, but have not considered this as a process or firmly established how the transition in social identity occurs following diagnosis. A review by Hole et al., [13] provided the greatest focus on social identity but did not consider: (a) the processes of how social identity evolves following diagnosis; or (b) the factors which may impact on it. A more comprehensive review is required to achieve this. Given the above, the aim of this study is to undertake a review of literature to discover a process of social identity evolution for people with stroke and identify the factors which influence it. 

## 2. Methods

A thematic synthesis approach to review was undertaken [14]. Search processes and outputs followed the Preferred Reporting Items for Systematic Reviews and Meta-Analyses (PRISMA) flow diagram and checklist [15]. A subtle realist viewpoint was assumed for this research. This philosophical view supports the idea of a common reality which can be established by considering the unique viewpoints given by individuals. This review provides an illustration of the common viewpoints through the thematic analysis whilst using quotes to identify the unique perspectives of individuals. Additionally, it has been argued that this view allows for a greater understanding of ethical, cultural and environmental factors within the qualitative data [16].

### 2.1. Protocol and Registration 

A protocol was registered with PROSPERO with the ID: CRD42019126481.

### 2.2. Eligibility Criteria

The PICOS (Participants, Intervention, Comparator, Outcome and Study design) acronym has been used to identify criteria of inclusion. PICOS was selected because of its high sensitivity in capturing qualitative data [17].

#### 2.2.1. P

The participants must have been clinically diagnosed with stroke a minimum of 6 months previously. Only studies with participants over the age of 18 years were included and perinatal stroke studies were excluded. This was because perinatal patients would have always had their condition and <6 months diagnosis of stroke is too soon to assess change over time [18]. Where multiple conditions were identified within a study, only results from stroke patients were included. If results included other populations, specific results on people with stroke had to be presented and at least 50% of the resulting data had to come from the participants who had suffered stroke. 

#### 2.2.2. I

Studies were only included if they focused on the experience of living with stroke using various types of qualitative techniques. Studies were required to identify processes involved with social identity development or factors which could influence the development of social identity. Experimental studies and mixed methods studies using intervention were excluded if they did not provide this detail. 

#### 2.2.3. C

Studies were included if a qualitative analysis on the meaning and experience of stroke was incorporated. 

#### 2.2.4. O

Any form of data collection which captured qualitative experiences relating to feeling and behaviours about living with a stroke were included. There was a need to capture experiences of living in order to capture the factors which influence change within the stroke patients’ experiences. 

#### 2.2.5. S

All qualitative methodologies such as interpretative phenomenological analysis (IPA), hermeneutic phenomenology, social constructivist grounded theory and ethnography were included. Case studies and qualitative fiction were excluded due to the aims of the research. All quantitative research was excluded due to the relativist ontology. For the purpose of this study other meta-synthesis or systematic reviews were not included within the literature synthesis due to their design. Studies undertaken within inpatient stroke services or outpatient settings were acceptable. 

### 2.3. Other Criteria 

In order to capture and include recent experiences of rehabilitation, an iterative process of searching and identifying studies was begun from 2010.

### 2.4. The Search Process 

A sensitive topic-based, iterative search strategy of various electronic databases (from 2009 until 1st January 2019) was undertaken including; MEDLINE, Child Development & Adolescent Studies, CINAHL Plus, eBook Collection (EBSCOhost), SPORTDiscus, EMBASE, PEDro, ZETOC databases. Specific internet resource sites were used to supplement the search including PubMed and Google Scholar (first 20 pages). The search terms included the following restrictions: language restrictions (English only), Humans, Peer reviewed and full text available from January 2009-2019. The following key words were identified: (stroke OR cerebrovascular accident OR CVA) AND (Narrative or Narratives, Storytelling OR story telling OR stories) AND (social identity OR self) AND (Factors AND experiences and perceptions) AND Intervention. Further searching of the five most common journals and three most common authors from the initial electronic search was undertaken and citation chasing was pursued. 

### 2.5. Study Selection 

The primary author undertook all searches and removed duplicates. This was followed by consideration of articles by title, abstract and full text. The selected articles were presented to the secondary author and exclusions were made in accordance with the eligibility criteria. 

### 2.6. Data Collection 

The primary author used a predefined extraction form as a guide to identify critical study-design information and demographical details from the chosen studies which was then systematically inputted into the form. The information collated into this table included: study name, methodology, method, participant details (age, gender, time post-stroke), sample method, geographical setting and aim of the study. This also included details of major themes, sub-themes and recommendations from the studies.

### 2.7. Study Quality Assessment 

The primary author assessed the quality of the qualitative studies using a modified 13-item version of the COREQ checklist [19]. This version removes items considered less sensitive within the 32-item version [20]. This fits with the paradigmatic position of the author. The quality assessment was undertaken to identify any fatally flawed studies (studies that are so poor that the data they present is questionable). The quality appraisal serves no further role for the purpose of this review.

### 2.8. Synthesis 

A general guide was followed using three stages according to proposed methods of synthesis. Stage 1 was *Open coding*. Open coding was undertaken by two authors and codes of interest were identified. Stage 2 was *Developing descriptive themes.* Descriptive themes were developed using two stages of mind mapping. This required consideration and reconsideration of topics. Common themes and sub themes from the raw data were extracted for the selected studies result sections [21]. Finally, Stage 3 was *Generating analytical themes and going beyond the data.* Going beyond the data is an important aspect for this type of qualitative synthesis [22]. In order to go beyond the data a process of identifying associations between the themes was undertaken because of their close relationship. This process can be seen in the Appendix A. After a full thematic analysis was identified, the final stage was to present it within a model related to social identity. For a full audit trail of the synthesis process, see the Appendix A for each stage.

### 2.9. Trustworthiness

The ENTREQ [23] was used as a reporting guideline for this review. The supervising author acted to undertake blind analysis as well as being a critical friend in regard to the analysis identified. 

## 3. Results

From the nine-hundred and fourteen studies screened, eighteen [7,24,25,26,27,28,29,30,31,32,33,34,35,36,37,38,39,40] were eligible for inclusion within this study. As seen below, Figure 1 presents the PRISMA flow diagram detailing the full search process. 

### 3.1. Demographics 

From the studies included within the qualitative synthesis, the total number of post-stroke participants was 251 (141/251, 56% male, 109/251, 43% female, 1 undisclosed). Most studies did not disclose the type of stroke diagnosis (13/18, 72%) with only five (5/18, 28%) specifying the diagnosis (11/251, 4.4% haemorrhagic; 35/251, 14% ischemic; 4/251, 1.5% CVA; 2/251, 0.5% multiple TIA’s; 199/251, 79% Not stated). Participants’ age ranged from 20–80+ years, with the average being 47–75 years old. All studies included length of time since stroke onset with; eleven studies stating > 6 months–2 years, six studies averages ranged 2–10.5 years and one study [36] stated the participant was 20 years post-onset. The most prolific setting for the interview data collection occurred at participants homes (n = 7). Other venues included: Local outpatients (n = 3), local University (n = 1), participants preference (n = 3) and not stated (n = 3). A plethora of countries were included with the UK being more common (n = 6). Other countries included: USA (n = 1), Denmark (n = 1), Indonesia (n = 1), Netherlands (n = 2), Canada (n = 2), Australia (n = 2) and Norway (n = 1). One study [27] was undertaken using internet-based data collecting and another did not state a location [36], but due to their author and journal affiliation, it is likely that they were based in the USA and Greece.

### 3.2. Quality Appraisal

Figure 2. displays article quality assessment findings as identified by the COREQ. The results from the study comparison COREQ [20] checklist found the highest scoring study to be Brown et al [25] achieving a total of 13/13. A between study comparison found that three studies scored 8/13 [24,26,33], and the lowest scoring two studies scored 4/13 [30,34]. No study was removed due to a fatal flaw. The average scores across study was 6.4/13. The weakest domain was domain 1 (2.1/5 42%), whilst the strongest was domain 3 (1.8/3 60%). The most frequent question which identified a low score was question 5 with 3/18 (16.7%) studies answering it. The Appendix A contains further evidence for the full consideration of the quality assessment.

### 3.3. Synthesis

The synthesis identified four major themes including (a) social identity, (b) risk taking (c) achievement of meaningful activities and (d) factors that can influence (barrier and facilitators) social identity development. The Appendix A provides a full explanation of these. 

The first three themes were associated together by a cycle suggesting that risk taking and achievement of meaningful activities and interaction can represent key aspects which help social identity evolve following a stroke. However, central biopsychosocial barriers could act to prevent this. A number system was used from 1–18 for the synthesis of the papers due to volume of data. The Appendix A can be used to source further quotes for each section. We have presented the major themes within a model (see Figure 3). The model is called the Evolution of Self-Identity Model. This model is devised of 5 stages which result in social identity evolution including; Stage 1 Present Social Identity, Stage 2 Risk Taking, Stage 3 Achievement of Meaningful Activities or Interactions, Stage 4 Re-evaluation of Social Identity and Stage 5 Future Social Identity. However, this cycle may be interrupted by some specific factors (barriers and facilitators) related to the processes involved. Below the model is presented as a process and detailed according to the given themes.

#### 3.3.1. Stage 1 Present Social Identity:

The present social identity was defined as the individuals’ personal understanding of their social identities and current adjustment to aspects of life influenced by the stroke. This was influenced in several ways: (a) by what can be achieved. Individuals had to consider times when their social identity and role within the community or family could change. This included what they could not achieve. Examples included the ability to work in a religious role supporting others [36] or undertaking household tasks [28,36]; (b) the role within the family or work identity. The onset of a stroke could change relationships. For instance, there were many negative experiences regarding the impact of the stroke on the family. Study [32] stated “*The stroke was destructive, my daughter moved out at age 16. I think she wanted to be far away*”. Alternatively, individuals who could not return to work lost an association with that social identity [34,39].

#### 3.3.2. Stage 2: Future Social Identity

In order to establish future social identity(ies), it appeared important that individuals understood that progression was gradual. The ability to achieve future social identities appeared dependent on several factors: (a) completing or adapting old activities. Once individuals understood that they may not get better and could accept the need to adapt [30] and discuss finding new hobbies new social identities were possible. For instance, one participant stated this as “*doing what you can, accepting what you cannot*”. Acceptance was identified as helping reduce anger and frustration resulting in increased quality of life [25]; (b) gaining a sense of purpose and pride. Individuals required the development of a sense of purpose in new responsibilities acquired. For instance, study [24] and study [26] both identified that caring for others allowed for a sense of accomplishment, making individuals feel capable. Another study [32] found being able to care for children gave pride in the social identity of being a parent. Furthermore, one participant explained “*I need to be better for my family ‘cause they need me”* [33]. This suggests that working towards a sense of purpose may increase motivation for recovery; (c) careful planning and preparation. Planning was identified as assisting the ability to achieve an adapted social identity. A participant from study [35] stated “*When shopping, I always prepare a shopping list, I write my list based on the route I take through the shop*”. However, a lack of spontaneity could have a perceived negative impact on individuals. For instance, an individual from study [32] discusses strains on their relationship by stating *“After some time, the changes in marital life became routine*”; (d) awareness of how you respond to progression. Individuals identified different ways to respond to progression or a lack of it. Responses were often as a positive recipient or as someone who became frustrated. Individuals identified the importance of recording and remembering progress previously. Study [24] explained “*As she was able to resume doll making, in her mind, she was not stuck at one place in time but rather was working along an ever-changing continuum toward a better quality of life*”. 

#### 3.3.3. Stage 3: Risk Taking

Risk taking was considered as a trial and error approach, which was identified as a way to take a step toward positive psychological adjustment and independence [34]. Risk taking was identified by three factors which made the ability to undertake tasks, activities and interactions more possible: (a) there was an understanding that making no attempt to change will result in no benefit or could make you worse off [39]. An individual in study [25] recognised that “*The worst thing you could do is sit around, you’re going downhill*!” In contrast, an individual from study [39] stated “*I have to take risks so I can get further*”; (b) there was a need to adjust to the environment and enable access to meaningful social roles, activities and interactions. A period of transition could be achieved by adapting and attempting something different and taking risks may assist with transition to successful living [35]; (c) continual defiance of circumstances was required. For instance, an individual from study [35] stated “*It’s a process of endless tinkering, weighing, adjusting, and coordinating mobility practices in each situation anew*”. For this individual, it was important to constantly adjust and take further risks to realise capabilities. This process was required to find their limitations. Even with high levels of impairment, some individuals would still take risks and manage to orchestrate support around them [39]. Alternatively, others identified participating in family activities as part of “*controlled adventuring*” [37].

#### 3.3.4. Stage 4: Achievement of Meaningful Activities or Interactions

One of the main goals of risk taking was to accomplish meaningful activities and interactions. This was achieved by individuals in three primary ways: (a) Activities undertaken must have a purpose for the individual, as unengaging tedious tasks may be counteractive to healthy living [25,34]; (b) Achieving meaningful activities or interactions provides recognised success and individuals gain confidence to tackle new challenges. This also allows further independence [7,34]. Overcoming challenges allowed for the question “*what’s next?*” [34] and gave the individuals the insight into a potential future, with one stating: *“I always try to do what I can within reason, I enjoy the challenge*” [7]; (c) By doing things autonomously, individuals gain a sense of purpose and begin to develop an evolved social identity. Study [39] states “*We made a list with different household activities I could do. I was really happy at the end of the day. I’d done all the household activities by myself, in my own way and own pace*”. This example described the evolution from being a stroke survivor to independent achievements to gain a sense of not being a burden. However, adapting to independence may be difficult, with some stating they felt “*vulnerable*” and “*feeling insecure*” when accommodating their limitations during daily life [30]. 

#### 3.3.5. Stage 5: Re-evaluation of Social Identity

The re-evaluation of social identity reflected the ability to reflect and plan on present circumstances. It was dependent on several aspects including: (a) Goal setting and progress. Individuals could often benefit from setting and measuring goals [33,40]. This could be crucial to the progression of their social identity. For instance, an individual in study [25] stated “*I got the job, I accomplished something that I didn’t think I’d be able to do*”. This accomplishment could reinforce a sense of hope; (b) Self-reflection of social identities and re-thinking priorities. Many individuals identified the need of establishing what is important in their life following a stroke and to accept new and evolving social identities as a result. One individual stated “*It’s as if I had one life and it ended when I had the stroke, and I now have a completely new life*” [28]. Study [34] discusses a strategy of looking back, around and forward to facilitate a consideration of all aspects of life and reworking self-image. The reflection of self appeared to allow the individual to re-evaluate their priorities, which may be different to those of their previous identity [27,30].

#### 3.3.6. Factors that can Influence Social Identity Evolution

The three factors which influence change include sub-themes: Social, Physical and Internal. Each sub-theme may facilitate or be a barrier towards social identity evolution. 

The social factors included: (a) Family support systems. People with stroke identified that on the one hand family support could provide access to more activities of daily living and could increase motivation to engage with meaningful activities and interactions. However, it could also leave people feeling that they had lost their autonomy and independence due to the interference of others. For instance, a participant in study [33] stated: “*they wrap you in cotton wool, last thing I wanted is to be an invalid”*. Thus, if family support becomes too over-protective, it may prevent progress for the individual by not allowing them to attempt activities; (b) Accessing meaningful social relationships. This included visiting old friends or work colleagues [29] this could be linked to improved mental well-being; (c) Achievements during rehabilitation and support. Individuals identified that therapy could provide them with a sense of progression [35]. However, some believed that therapists could not help them [7,29,30]. Post discharge when support from therapists was reduced, individuals could become more unsure of what was possible. For instance, one study [31] stated that individuals “*felt it left a gap or uncertainty of further improvements*”; (c) Social isolation and inclusion. Loss of communication with others could isolate individuals [25,27,32]. For instance, study [25] stated *“Even if it’s friends coming here. You don’t know how to (indicates to mouth)”.* This was a definitive barrier to inclusion with social situations, increasing frustration during social interactions. However, humour was identified as one way to overcome such experiences [25].

The physical factors included: (a) The level of physical recovery achieved. Physical recovery impacts upon individuals differently with some regaining a sense of self-confidence and feeling human [24,31,33]. Others felt more physically able to undertake activities within the family [7,27,29]. Some individuals appeared to be influenced by their perception of change to masculinity or feeling feminine [29,30]. It was stated by one individual that “*You lose hope because you can’t get better”* [7]; (b) Mobility status. Mobility status allowed difference perceived benefits for individuals. For instance, one individual stated *“if we travel to the fishing village, we could see our son*” [39]. For others, it allowed independence and freedom. This had the opposite effect when mobility was not achieved [7,25,33]. One individual stated *“Because I can’t get out there and do what I want. So—just got to stay here to do nothing*” [25].

The internal factors included: (a) Acceptance of their condition. Acceptance appeared to allow individuals to be open to moving forward due to positive attitude towards adapting activities [7,25,40]. Inability to accept the condition and change could mean individuals became fixated on trying to achieve a pre-stroke level of function or stuck at their present perceived social identity. This could cause anger and a feeling of vulnerability towards the future [34]; (b) Motivation. Studies identified that “having a positive attitude” [25] and the willingness to engage in “doing things” [2,17,18] helped individuals have an emotional shift from negativity to positivity. This facilitated hope, leading to the factor of; (c) Hope in possibility. The ability to consider a past, present and future social identity could be met with contrasting reactions from denial towards hope. This was mostly influenced by the ability to accept the potential change. Individuals were more positive when they could see the possibility of a desired future while accepting that it may not come about. However, this view was often not possible. For instance, one individual stated “*There’s more to me than what you see. I was such a vibrant person, Now there’s nothing*.” [24]. Such expressions could be followed by anger and vulnerability for the future [34].

## 4. Discussion

Within literature, there has been some discussion over the psychosocial processes involved with reclaiming social identity post-stroke. This study presents a thematic synthesis of qualitative research, reviewing the most common themes associated within the process that illustrates how social identity evolves or is hindered post-stroke. Physical, internal and social barriers act to prevent the evolution of an individual’s social identity. If these barriers can be overcome, it is more likely that individuals will have the confidence to take risks by attempting meaningful activities and begin a process of re-evaluating social identity. However, the inability to psychologically adapt and accept an altered future social identity can mean individuals experience further frustration, anger and isolation [25]. The current results demonstrated that risk taking allowed stroke survivors to recognize their progress and helped to evolve their social identity. It also provided a sense of pride and purpose because of the activities that become possible and the roles that can be accessed. Other similar studies [41,42] considering the journey of people with stroke have recognized social identity as a key factor in enhancing quality of life. Past research [13,41] has found themes related to reclaiming social identity included: agency, control, disease concept and doing things as factors which positively impacted on mental health. The current research supports these findings.

A meta-ethnography study by Hole et al. [13] developed a linear model where hope, social support and self-efficacy are influenced by clinical staff and external support. These social factors should be identified when considering elements which can impact, and even impede upon social identity development. Sharing different stories and views about evolution of living with stroke may be the best way to evolve social identity and improve social support [43]. The findings from the current review would support this, as sharing stories would include overcoming barriers and give rise to multiple different possibilities in the future for the listener. 

Further to this, healthcare professionals (HCPs) need to recognise that an individual’s social identity can be defined by the story they tell [44] and HCPs need to develop an understanding of the importance of capturing meaningful content from the patient’s stories. Listening, support and gaining trust between the patient and therapist may allow access to identity through the stories told and where their recovery process is being interrupted [45]. Considering what is purposeful and has meaning to them while identifying attractor patterns may influence their drive and motivation to change [46]. Using negative stories are important, as these highlight what is inhibiting their desired identity and would aid transition to Stage 3 of the evolution of social identity model.

### 4.1. Clinical Implications 

Using the acronym “SIP” is memorable for HCPs to implement this model. This encompasses the three sub-themes: Social (S), Internal (I) and Physical (P) factors which influence acceptance and risk taking. (a) Social support (S) within the community and family settings should be encouraged, as should access to others who have taken risks through storytelling intervention [47]. Family support may facilitate safe, achievable activities into the daily routine to increase independence [32]. (b) Internal (I) psychological adaptation is accessed by proving hope in possibility, recognising where they were to where they are now by setting realistic measurable goals [24]. This may lead the patient to be more motivated and guide them towards acceptance. (c) Physical recovery (P) during rehabilitation ensures patient choice and that the session targets developing meaningful activities [30]. This process can be complex as the individual needs to re-evaluate and re-think priorities; the rehabilitation should mirror this [25].

The simplest application for supporting the SIP principles in clinical practice could be achieved using an adapted hope and adaptation scale developed for neurological populations [48] (see Appendix B). The adapted scale contains three questions. The first question requires the individual to self-identity a meaningful activity, work role or relationship that has been impacted following the diagnosis and which has influenced their psychological adaptation the most. The following two questions use a 9-point Likert scale to determine how hope and psychological adaptation have been influenced. Based on this response, the clinician will be able to determine a recommendation to enhance or re-establish social identity. Based on previous research [48], a negative score on either the hope or adaptation question would result in the clinician responding in a more supportive way; this response would focus on observation to consider how others have re-established meaningful activities, interactions and social identities following a stroke. A positive score on either scale would require the clinician to support and encourage further connections and engagement with the meaningful activity identified or others.

### 4.2. Limitations

This model is a new way to view the evolution of identity. Although evidence-based research has been synthesised and quality appraisal assured, there is no guarantee of its success in practice. The included studies use different eligibility criteria and the included sample may be limited by this. Some interviews within studies synthesised were undertaken with a partner present (3/18, 16%) which may have impacted on the honesty or influenced the view of the patient’s story [35,36,37]. However, due to the small number of studies using this style of intervention, this would not have impacted greatly on the results. The data must be considered with caution when you consider the different healthcare environments and cultures from which the data was gathered. The data should not be considered as reflecting a single truth, instead reflecting a common reality which has some representativeness with each stroke cohort.

## 5. Conclusions

The current meta-synthesis has highlighted the major themes within the evolution of identity and management strategies for risk taking to achieve a desired future. The Evolution of the Self-Identity Model has been designed to reflect the complex nature of the recovery process. It is important for HCPs to use the principles from this model to engage with patients. Further research is required to determine the benefits of this model.

## Figures and Tables

**Figure 1 behavsci-10-00046-f001:**
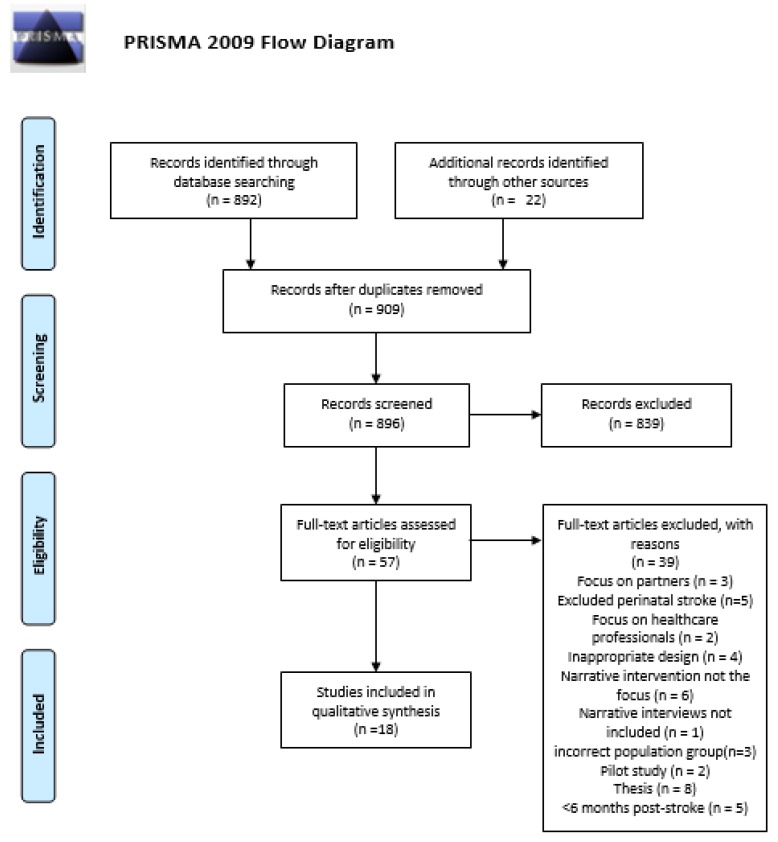
The Preferred Reporting of Items for Systematic Reviews and Meta-Analysis (PRISMA) Flow Diagram.

**Figure 2 behavsci-10-00046-f002:**
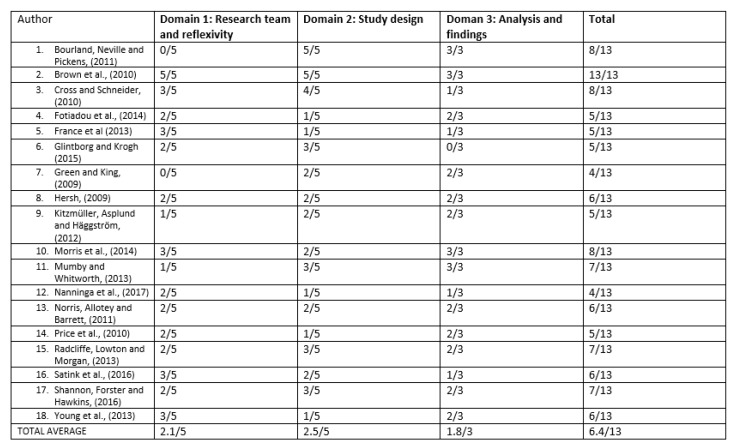
Modified 13-item COREQ checklist [19].

**Figure 3 behavsci-10-00046-f003:**
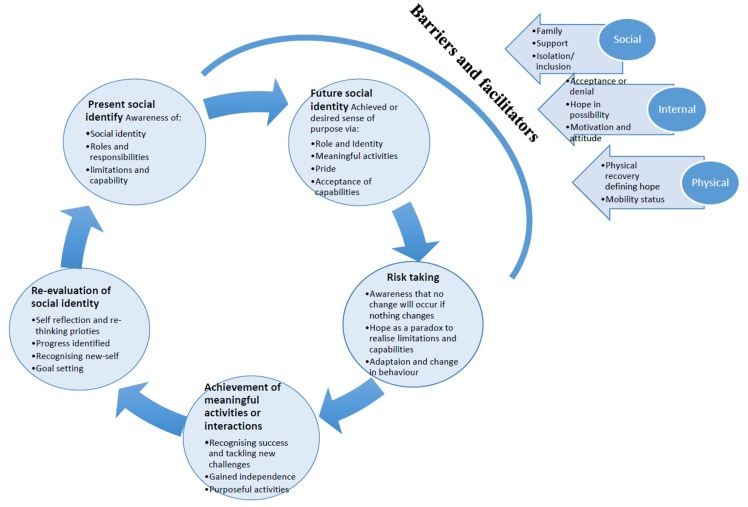
The Evolution of Self-Identity Model.

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
