# Peer review of "Factors which Influence Risk Taking and the Evolution of Social-Identity in Stroke Narratives: A Thematic Synthesis"

_behavsci, 2020, doi:10.3390/bs10020046_

Round 1

Reviewer 1 Report

This is a well-written and proper conduct meta-ethnographic systematic review to construct the major themes, sub-themes, and codes to determine the processes model and factors that influence the evolution of social identity for people with a stroke. If possible, Please re-consider the relationship and arrangement of the Evolution of Self-Identity Model in order to more closely explain and answer the research quesitons.

Author Response

Reviewer 1

This is a well-written and proper conduct meta-ethnographic systematic review to construct the major themes, sub-themes, and codes to determine the processes model and factors that influence the evolution of social identity for people with a stroke.

AS: thank you for these comments.

If possible, Please re-consider the relationship and arrangement of the Evolution of Self-Identity Model in order to more closely explain and answer the research questions.

AS: thank you for these comments. Reviewer two also highlights the need to explain how the model was developed. We have reorganised the supplementary file and presented the results according to the model.

Reviewer 2 Report

Thank you for the opportunity to review this paper: a meta-ethnography on social identity in stroke narratives. 

I thought the search and selection processes was well conducted. My main issue with the paper was the use of the term 'meta-ethnography' as I'm not sure that the authors have followed the steps of a meta-ethnography to analyse and synthesise the included papers. I was expecting to see that the authors had followed the specific 7-step meta-ethnographic approach as described by Noblit and Hare, and include such steps as determining how the studies are related, translating the studies into one another, and producing a line of argument. I wasn't sure if the model the authors produced (section 3.8 in the results) was the 'line of argument'. I wasn't sure if the model was derived from the synthesis of the 18 papers - the results of which are reported in sections 3.4-3.7. So maybe the authors could provide some detail on how they got from sections 3.4-3.7 to the model in 3.8. 

I found the results quite hard to read - there's a lot to take in for the reader  - perhaps try changing the structure of the sentences as I found it a bit repetitive to read with many of the paragraphs starting in the same way '  'this sub-theme was defined as', 'this code was defined as'.

There are a few grammatical errors, typos, words missing in sentences so the paper could do with a check through. 

Author Response

Thank you for the opportunity to review this paper: a meta-ethnography on social identity in stroke narratives. 

I thought the search and selection processes was well conducted. My main issue with the paper was the use of the term 'meta-ethnography' as I'm not sure that the authors have followed the steps of a meta-ethnography to analyse and synthesise the included papers. I was expecting to see that the authors had followed the specific 7-step meta-ethnographic approach as described by Noblit and Hare, and include such steps as determining how the studies are related, translating the studies into one another, and producing a line of argument.

AS: We note the reference to Noblit and Hare 1988 and agree. Since that an in the early 2000 onwards there were some examples of a move for some meta-ethnography’s to be presented in three stages. However, I (AS) was aware of the new emerge (meta ethnography) guidelines (France et al., 2019). We accessed this document and compared against the included the supplementary file which we submitted. This illustrated that the processes that were undertaken were akin to thematic synthesis which represents what we have done exactly. Thank you for bringing this point to our attention.

I wasn't sure if the model the authors produced (section 3.8 in the results) was the 'line of argument'. I wasn't sure if the model was derived from the synthesis of the 18 papers - the results of which are reported in sections 3.4-3.7. So maybe the authors could provide some detail on how they got from sections 3.4-3.7 to the model in 3.8. 

AS: Interestingly the thematic synthesis paper mentions this idea and identifies the importance of going beyond the data and uses references including ideas from meta-ethnography. So this fits with the review type identified. We have added the detail into explain how we went from the themes to the module.  

I found the results quite hard to read - there's a lot to take in for the reader  - perhaps try changing the structure of the sentences as I found it a bit repetitive to read with many of the paragraphs starting in the same way '  'this sub-theme was defined as', 'this code was defined as'.

AS: We have reorganised the results to bring them into the model to show the that the model and the themes were closely related.

There are a few grammatical errors, typos, words missing in sentences so the paper could do with a check through. 

AS: these have been checked. thanks

Round 2

Reviewer 2 Report

I think the paper has improved enormously. The results section, in particular, is so much easier to read and take in. I think it was a really good decision to change the method of synthesis to thematic synthesis rather than meta-ethnography, and you still have been able to produce a result that is greater than the sum of the parts, going beyond what is reported in the included articles. The paper is now a really engaging read.

There are still a few typos or sentences which don't read quite right. I tried to highlight them in the attached document but it's not necessarily comprehensive. Note that there's still a meta-ethnography term left in the conclusion section. 

Author Response

I think the paper has improved enormously. The results section, in particular, is so much easier to read and take in. I think it was a really good decision to change the method of synthesis to thematic synthesis rather than meta-ethnography, and you still have been able to produce a result that is greater than the sum of the parts, going beyond what is reported in the included articles. The paper is now a really engaging read.

AS: Thank you for your support and suggestions to make this possible.

There are still a few typos or sentences which don't read quite right. I tried to highlight them in the attached document but it's not necessarily comprehensive. Note that there's still a meta-ethnography term left in the conclusion section. 

AS: I have had made changes as suggested and also added further changes.